# IFN-mediated negative feedback supports bacteria class-specific macrophage inflammatory responses

Rachel A Gottschalk[1†‡*], Michael G Dorrington[2†], Bhaskar Dutta[1§], Kathleen S Krauss[1], Andrew J Martins[3], Stefan Uderhardt[1], Waipan Chan[1], John S Tsang[3], Parizad Torabi-Parizi[4], Iain DC Fraser[2], Ronald N Germain[1*]

[1]Lymphocyte Biology Section, Laboratory of Immune System Biology, National Institute of Allergy and Infectious Diseases, National Institutes of Health, Bethesda, United States; [2]Signaling Systems Section, Laboratory of Immune System Biology, National Institute of Allergy and Infectious Diseases, National Institutes of Health, Bethesda, United States; [3]Systems Genomics and Bioinformatics Unit, Laboratory of Immune System Biology, National Institute of Allergy and Infectious Diseases, National Institutes of Health, Bethesda, United States; [4]Critical Care Medicine Department, Clinical Center, National Institutes of Health, Bethesda, United States

**\*For correspondence:**
rachel.gottschalk@pitt.edu (RAG);
rgermain@niaid.nih.gov (RNG)

[†]These authors contributed equally to this work

**Present address:** [‡]Department of Immunology, School of Medicine, University of Pittsburgh, Pittsburgh, United States; [§]Oncology R&D, AstraZeneca, Gaithersburg, United States

**Abstract** Despite existing evidence for tuning of innate immunity to different classes of bacteria, the molecular mechanisms used by macrophages to tailor inflammatory responses to specific pathogens remain incompletely defined. By stimulating mouse macrophages with a titration matrix of TLR ligand pairs, we identified distinct stimulus requirements for activating and inhibitory events that evoked diverse cytokine production dynamics. These regulatory events were linked to patterns of inflammatory responses that distinguished between Gram-positive and Gram-negative bacteria, both in vitro and after in vivo lung infection. Stimulation beyond a TLR4 threshold and Gram-negative bacteria-induced responses were characterized by a rapid type I IFN-dependent decline in inflammatory cytokine production, independent of IL-10, whereas inflammatory responses to Gram-positive species were more sustained due to the absence of this IFN-dependent regulation. Thus, disparate triggering of a cytokine negative feedback loop promotes tuning of macrophage responses in a bacteria class-specific manner and provides context-dependent regulation of inflammation dynamics.
DOI: https://doi.org/10.7554/eLife.46836.001

## Introduction

When host cells come into contact with microbes, they receive a complex set of signals based on the biochemical composition of the foreign entity. The combination and timing of receptor ligation by microbial products encodes pathogen identity, providing the host cell with the information needed to produce an appropriate response, both in form and amplitude (*Takeuchi and Akira, 2010*). Indeed, the class of bacterial pathogen has been correlated with inflammatory response magnitude during in vitro macrophage stimulation and with distinct cytokine profiles in human disease (*Abe et al., 2010*; *Hessle et al., 2005*; *Surbatovic et al., 2015*). Too much inflammation results in extensive host tissue damage, while too little allows the pathogen to replicate and overcome the host's natural processes (*Bachmann and Kopf, 2002*; *Nimmerjahn and Ravetch, 2008*). Thus, appropriate tuning of innate inflammatory responses is integral to the survival of higher organisms as they interact with microbes throughout their lives.

While the existence and importance of such pathogen-specific innate immune tuning is well-established, there are many gaps in our understanding of the molecular events that underlie such discriminatory responses. Bacterial pathogens are recognized by extracellular and intracellular pattern recognition receptors (PRRs), such as Toll-like receptors (TLRs), based on their expression of pathogen-associated molecular patterns (PAMPs)(*Mogensen, 2009*). Ligation of these receptors leads to signaling cascades in the macrophage that result in the transcription of various inflammatory genes including cytokines, chemokines, and transcription factors, as well as negative regulators of the transcriptional events needed for the elaboration of these mediators (*Lang and Mansell, 2007*). TLRs engage two major intracellular signaling pathways based on the primary adaptor proteins involved. Inflammatory responses are largely controlled by NF-kB and MAP kinases activated downstream of the myeloid differentiation primary response 88 (MyD88) adaptor, while the TIR-domain-containing adaptor-inducing interferon-beta (TRIF) pathway induces interferon-regulatory factor (IRF) activation and the production of type I interferon (IFN) (*O'Neill and Bowie, 2007*). While the thick outer coats of Gram-positive bacteria preferentially evoke MyD88-dependent signaling, Gram-negative bacteria like those belonging to the genera *Pseudomonas*, *Salmonella,* and *Escherichia* express high levels of lipopolysaccharide (LPS) anchored to the cell membrane, which activates both the MyD88 and TRIF pathways through engagement of TLR4 (*Poltorak et al., 1998*; *Ulevitch and Tobias, 1999*; *Yamamoto et al., 2003*).

We hypothesized that responses to complex combinations of ligands displayed by different classes of bacterial pathogens may reflect the operation of interacting feedback controls evoked by engagement of the MyD88 and/or TRIF pathways. By stimulating macrophages with a titration matrix of pure TLR ligands, here we show that the behavior of core inflammatory genes is governed by activating and inhibitory mechanisms downstream of distinct input requirements. This leads to TLR ligand and concentration-dependent inflammatory cytokine dynamics that explain dichotomous cellular responses to Gram-positive and Gram-negative bacteria. In particular, TLR4-dominant Gram-negative stimulation leads to a rapid type I IFN-mediated decline in production of select inflammatory cytokines, due to induction of a robust IL-10 independent negative feedback circuit. This disparate triggering of IFN-mediated negative feedback was responsible for quantitatively and temporally distinct inflammatory responses distinguishing macrophage exposure to major classes of bacteria. Pathogen-specific, IFN-dependent cytokine regulation was recapitulated in mouse models of *P. aeruginosa* and *S. aureus* lung infection. Together, we propose a model based on studies involving quantitative binary ligand stimulation and intact bacteria in which the relative levels of MyD88 and TRIF pathway activation are integrated to support context-dependent inflammatory control.

## Results

### TLR pathway non-additivity supports context-specific inflammatory responses

In an effort to systematically examine macrophage inflammatory responses to bacterial pathogens and identify events that could be linked to distinctive responses to broad classes of these organisms, we stimulated bone marrow-derived macrophages (BMDM) with a panel of heat-killed bacteria, including two Gram-positive species and three Gram-negative species (*Figure 1A*; red and blue legends, respectively). Supernatants were collected after 12 hr of stimulation and six inflammatory mediators were quantified using cytometric bead array. As expected, all five species of bacteria induced robust inflammatory cytokine and chemokine production. However, the resulting cytokine output varied, with significant differences in the nature and magnitude of the inflammatory mediators induced in response to Gram-positive versus Gram-negative species (*Figure 1A*). In particular, more of the chemokine CXCL1 and the cytokine TNF were found in the culture supernatant following macrophage exposure to Gram-positive species, whereas the chemokine CCL2 and the cytokine IL-6 accumulated in greater amounts in response to Gram-negative species.

To conduct more quantitative studies, we sought a reductionist system that recapitulated such bacteria class-specific inflammatory mediator production. Considering the key role of cell surface-associated TLRs in the initiation of responses to bacteria, we assessed responses to the TLR4 ligand Kdo2-LipidA (KLA; the stimulatory unit of LPS) and the TLR2/TLR1 ligand Pam3CSK4 (P3C; a synthetic bacterial lipoprotein). While LPS is found exclusively on Gram-negative species, bacterial

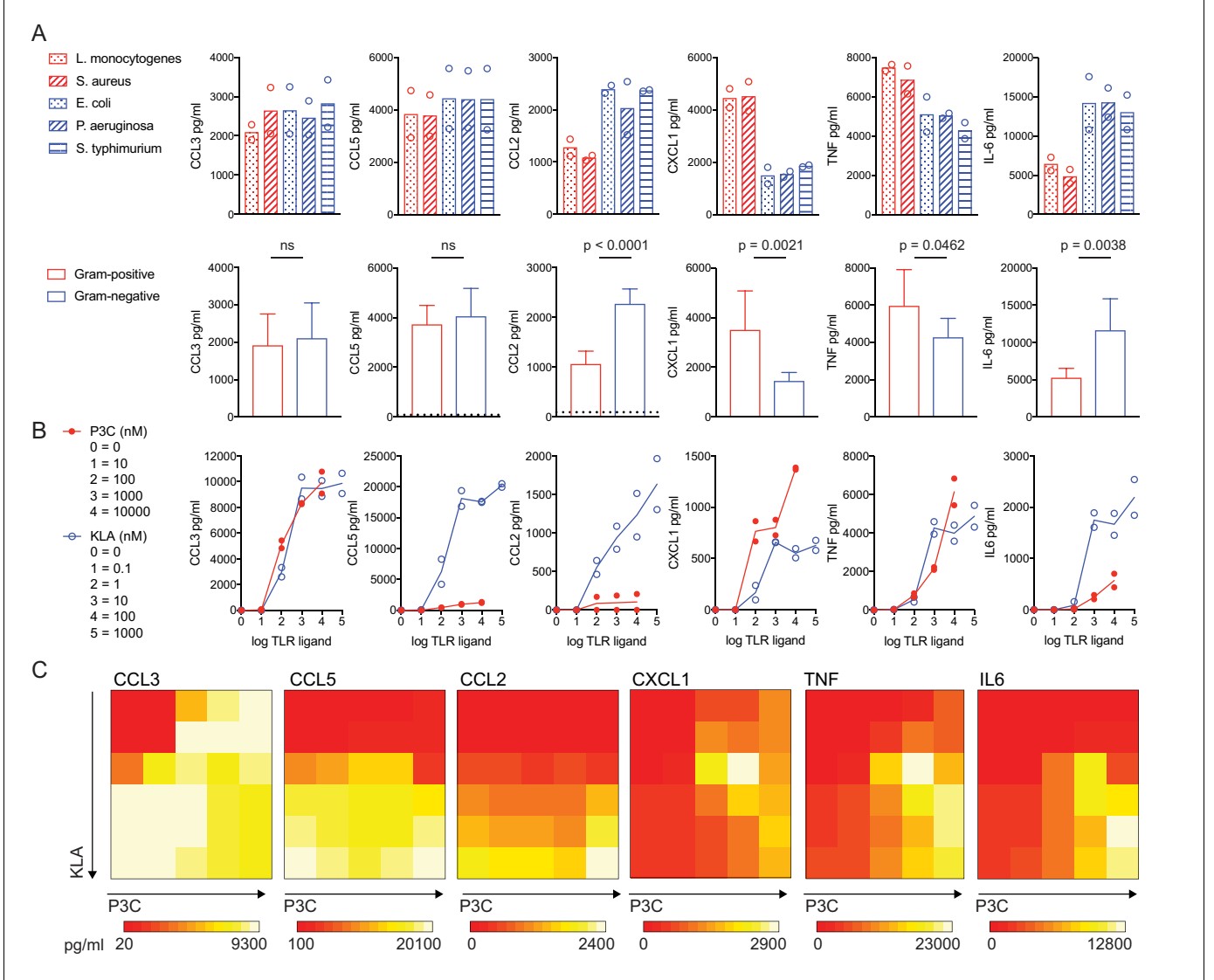

**Figure 1.** TLR pathway non-additivity supports context-specific inflammatory responses. BMDM were stimulated with heat-killed bacteria at an MOI of 100 (Red = Gram-positive: *L. monocytogenes* or *S. aureus*; Blue = Gram-negative: *E. coli*, *P. aeruginosa*, or *S. Typhimurium*) or soluble TLR ligand (P3C = TLR2 ligand Pam3CKS4; KLA = TLR4 ligand KDO2-LipidA), as indicated. After 12 hr (A) or 6 hr (B, C) supernatants were collected and cytokines were quantified using cytometric bead array. Data points represent independent experiments. (A) Data from bacterial species (upper row) were combined from three independent experiments to show the distinct effects of Gram-positive and Gram-negative bacteria on inflammatory cytokine production in BMDM (lower row); statistical significance based on unpaired t-tests, between Gram-negative- and Gram-positive-stimulated samples. Dotted lines in the lower row represent the mean values for unstimulated BMDM (0–12 hr). In some cases, these controls are not visible because they are below the level of detection (a zero value). Ligand concentrations in (C) correspond to single ligand concentrations shown in (B), in pairwise combinations. Data are representative of three independent experiments.

DOI: https://doi.org/10.7554/eLife.46836.002

The following figure supplements are available for figure 1:

**Figure supplement 1.** TLR2 and TLR4 ligation by heat-killed bacteria.

DOI: https://doi.org/10.7554/eLife.46836.003

**Figure supplement 2.** Linear regression analysis highlights non-additive TLR pathway interactions.

DOI: https://doi.org/10.7554/eLife.46836.004

lipoproteins that ligate TLR2 are produced by Gram-positive and Gram-negative bacteria (*Aliprantis et al., 1999*). We confirmed the expected bacterial ligation of TLR4 and TLR2 by stimulating HEK293 cells stably expressing these TLRs with our panel of heat-killed bacteria and assessing their production of IL-8 (*Figure 1—figure supplement 1*). In conducting these experiments with purified ligands, we utilized a dual ligand titration matrix approach (single ligand titrations and titration matrix, *Figure 1B and C*, respectively). We reasoned that this system would allow us to observe pathway interactions relevant to combinatorial stimulation in the context of macrophage discrimination of bacterial species, uncovering responses that would not be apparent upon stimulation with a single bacterial product. Indeed, in the case of CXCL1 and TNF, the dual ligand matrix revealed an unexpected topology, with peak cytokine production detected at a combination of concentrations that were suboptimal for either individual ligand (1 nM KLA + 100 nM and 1000 nM P3C, *Figure 1C*); we termed this region of the TLR2 by TLR4 titration matrix the 'hotspot'. In contrast, other inflammatory mediators appeared most robustly induced by the highest concentrations of individual or combined TLR ligands.

In an effort to better understand the pathway interactions leading to these distinct patterns of cytokine production, we used the same stimulation matrix and after two hours measured gene expression using microarray. We then used four linear regression models to examine the variation in the expression of each gene across these 30 conditions. In essence, these models interrogate how much variation in mRNA expression can be explained by the ligand concentration of TLR2 alone, TLR4 ligand alone, and the addition of TLR2 and TLR4 ligands together, while also evaluating the degree of non-additive interaction between the TLRs. Comparison of the adjusted $R^2$ values for the single TLR models suggested that TLR4 was a stronger determinant of gene expression in the context of dual TLR stimulation than TLR2/1 (*Figure 1—figure supplement 2*). However, for a portion of genes with low $R^2$ values for either single TLR model, the more complex interactive model captures more of the variation (*Figure 1—figure supplement 2*, yellow points below 0.5 $R^2$ on either axis). This interactive model outperforms the additive model for a proportion of TLR-induced genes, highlighting the non-linear nature of TLR pathway interactions (*Figure 1—figure supplement 2*).

## PAMP combination and concentration determine negative feedback control of inflammatory output

Given past evidence for synergistic behavior of TLR ligands in evoking specific cytokine responses from macrophages (*Lin et al., 2017*; *Napolitani et al., 2005*; *Tan et al., 2014*), we speculated that genes with products contributing to the non-additive regulation of CXCL1 and TNF production may be differentially expressed in the corresponding areas of the dual ligand titration matrix. To examine this hypothesis, we utilized our microarray data to identify context-dependent regulators, assessing fold changes in the hotspot, where maximum CXCL1 and TNF was detected, as compared to the TLR4-high region (*Figure 2A*). Surprisingly, very few genes were more robustly induced in the hotspot (*Figure 2B* and *Figure 2—figure supplement 1*). Instead, a larger number of genes showed increased expression in response to high concentrations of TLR4 ligand where the mediator output was diminished relative to the hotspot (*Figure 2B* and *Figure 2—figure supplement 1*). These findings led us to consider a different model in which high concentrations of TLR ligand(s) induced increased expression of negative regulators that dampened the responses seen at lower concentrations of ligand. More specifically, we postulated that some of the genes showing increased expression at high KLA concentrations might be directly or indirectly dampening CXCL1 and TNF production, and that decreased induction of these regulators at lower TLR ligand concentrations may explain the higher cytokine expression in the hotspot.

To test this hypothesis, seven candidate regulators were selected from the list of differentially expressed genes based on putative function or described regulatory activity in other experimental systems: *Nfil3* (*Cowell et al., 1992*), *Batf2* (*Muto et al., 2004*), *Themis2* (*Peirce et al., 2010*), *Cish* (*Palmer et al., 2015*), *Chd1* (*Gaspar-Maia et al., 2009*), *Rcan1* (*Fuentes et al., 2000*), and *Pcgf5* (*Yao et al., 2018*) (*Figure 2B*). Expression of these genes was decreased ('knocked down') in the macrophages through use of three distinct siRNA per gene and both non-targeting (NTC) and Myd88 siRNAs were used as controls. After siRNA knockdown, BMDM were stimulated with ligand conditions corresponding to the TLR4-high region of the titration matrix (1000 nM P3C and 100 nM KLA) and TNF was measured in the supernatant. Consistent with their having negative regulatory

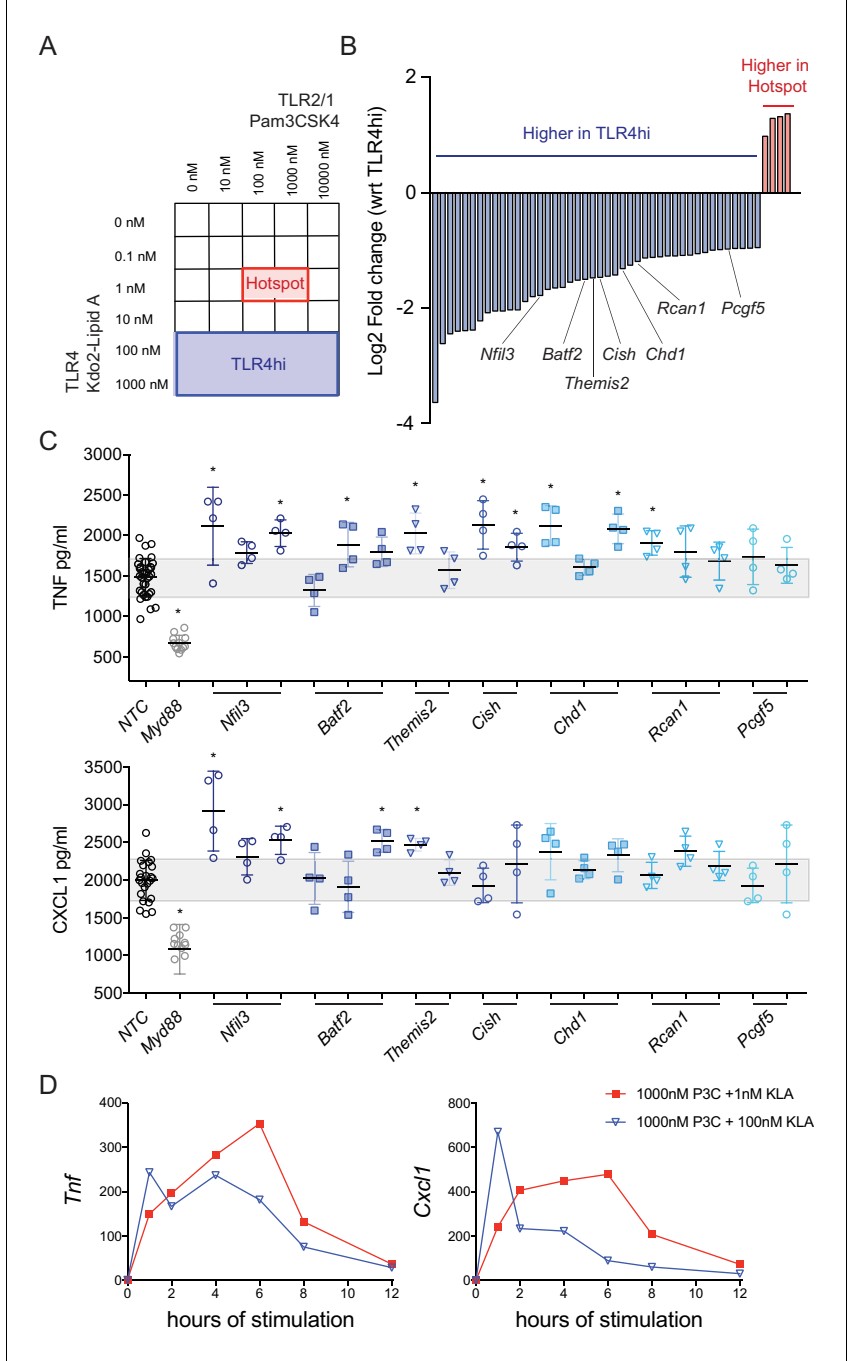

**Figure 2.** Regulator genes are induced above a threshold of TLR4 ligand. (A, B) Using the titration matrix conditions shown in *Figure 1*, BMDM were stimulated for 2 hr and induced gene expression was measured by microarray. Differential gene expression analysis was performed between the hotspot and TLR4 high regions of the dual ligand titration matrix. Candidate regulator genes preferentially induced by high TLR4 ligand are noted (B). (C) BMDM were treated with three independent siRNA per candidate regulator (siRNA resulting in less than 50% viability were excluded), MyD88 siRNA, or non-targeting control (NTC) siRNA for 48 hr prior to stimulation with 1000 nM P3C and 100 nM KLA (TLR4hi condition). After 6 hr of stimulation, TNF or CXCL1 was quantified in supernatants by ELISA and values were normalized based on cell viability. Individual data points represent wells treated independently with siRNA and TLR ligand, and the gray bar depicts NTC standard deviation. Candidate genes were differentially expressed in three independent microarray or qPCR experiments and siRNA results are representative of two independent experiments. Stars represent statistical significance based on ordinary one-way ANOVA, versus control (*p≤0.05). (D) BMDM were stimulated with the indicated concentrations of TLR ligand and

*Figure 2 continued on next page*

*Figure 2 continued*

harvested at the indicated time. Gene expression was quantified using qPCR and fold change was calculated compared to unstimulated BMDM. Data are representative of three independent experiments.

DOI: https://doi.org/10.7554/eLife.46836.005

The following source data and figure supplement are available for figure 2:

**Source data 1.** Microarray data for titration matrix.

DOI: https://doi.org/10.7554/eLife.46836.007

**Figure supplement 1.** Genes induced above a threshold of TLR4 activation.

DOI: https://doi.org/10.7554/eLife.46836.006

function, knockdown of six of these candidate genes resulted in increased TNF secretion, while three reached statistical significance for CXCL1 secretion, compared to the NTC siRNA (*Figure 2C*).

To examine whether this negative regulation acted immediately downstream of TLR engagement, prior to initial cytokine responses, or played a role in shutting down an early response, we measured *Tnf* and *Cxcl1* mRNA over time. We compared conditions corresponding to the hotspot (1000 nM P3C + 1 nM KLA) and TLR4-high (1000 nM P3C + 100 nM KLA) regions over 12 hr of stimulation (*Figure 2D*). High concentrations of KLA result in *Tnf* decline after four hours of stimulation with an even more rapid and sharp decline in *Cxcl1* after one hour of stimulation, consistent with strong TLR4-specific negative regulation. In contrast, ligand concentrations associated with the hotspot yielded a more sustained *Tnf* and *Cxcl1* response, with mRNA increasing through six hours. Together, these findings suggested that TLR-specific differences in the induction of negative feedback regulation might play an important role in shaping bacteria class-specific inflammatory responses.

## TLR ligand titration matrix analyses predict bacteria class-specific regulator induction and cytokine kinetics

The preceding experiments utilizing a soluble ligand titration matrix identified distinct negative feedback responses sensitive to dual TLR engagement conditions. We next sought to determine whether these findings provided direct mechanistic insight into the bacteria class-specific inflammatory responses we had observed (see *Figure 1*). While many bacteria class-specific inflammatory response differences were observed with multiple concentrations of heat-killed bacteria (*Figure 3—figure supplement 1*), we selected a concentration where the results were most consistent between groups (equivalent to MOI of 100). Using our panel of heat-killed bacteria, we stimulated BMDM for six hours before removing the supernatant and replacing the medium to capture cytokines produced late after stimulation. Considering the high expression of LPS by Gram-negative bacteria, we anticipated from the dual ligand matrix data that we would observe a dramatic decline of CXCL1 and TNF produced late in response to the three Gram-negative species. Indeed, we observed transient production of CXCL1 and TNF in response *E. coli*, *P. aeruginosa*, and *S. Typhimurium*, with minimal secretion of these cytokines after six hours (*Figure 3A*). In contrast, we observed sustained CXCL1 and TNF production in response to the Gram-positive bacteria *L. monocytogenes* and *S. aureus* (*Figure 3A*), as was seen in the hotspot of the dual ligand matrix. In addition to distinct cytokine production kinetics, expression of the negative regulators highlighted by our titration matrix analyses also showed bacteria class-specific expression. These genes were induced much more robustly in response to Gram-negative bacteria as compared to BMDM stimulated with Gram-positive bacteria (*Figure 3B*). Some degree of bacteria-specific induction was also observed for other known positive and negative regulators of inflammatory responses, though the magnitude of difference between bacterial classes was generally less than seen for the candidates identified using the dual TLR ligand matrix (*Figure 3—figure supplement 2*). Thus, PAMP-specific negative feedback supports context-dependent inflammatory responses.

## IFN-mediated negative feedback supports bacteria-specific responses

In seeking to understand the origin of the differing kinetics of responses to distinct ligand and ligand combinations, we considered the unique capacity of TLR4 to engage both the MyD88 and TRIF pathways, in contrast to TLR2 ligation that stimulates signaling only through MyD88. TRIF signaling

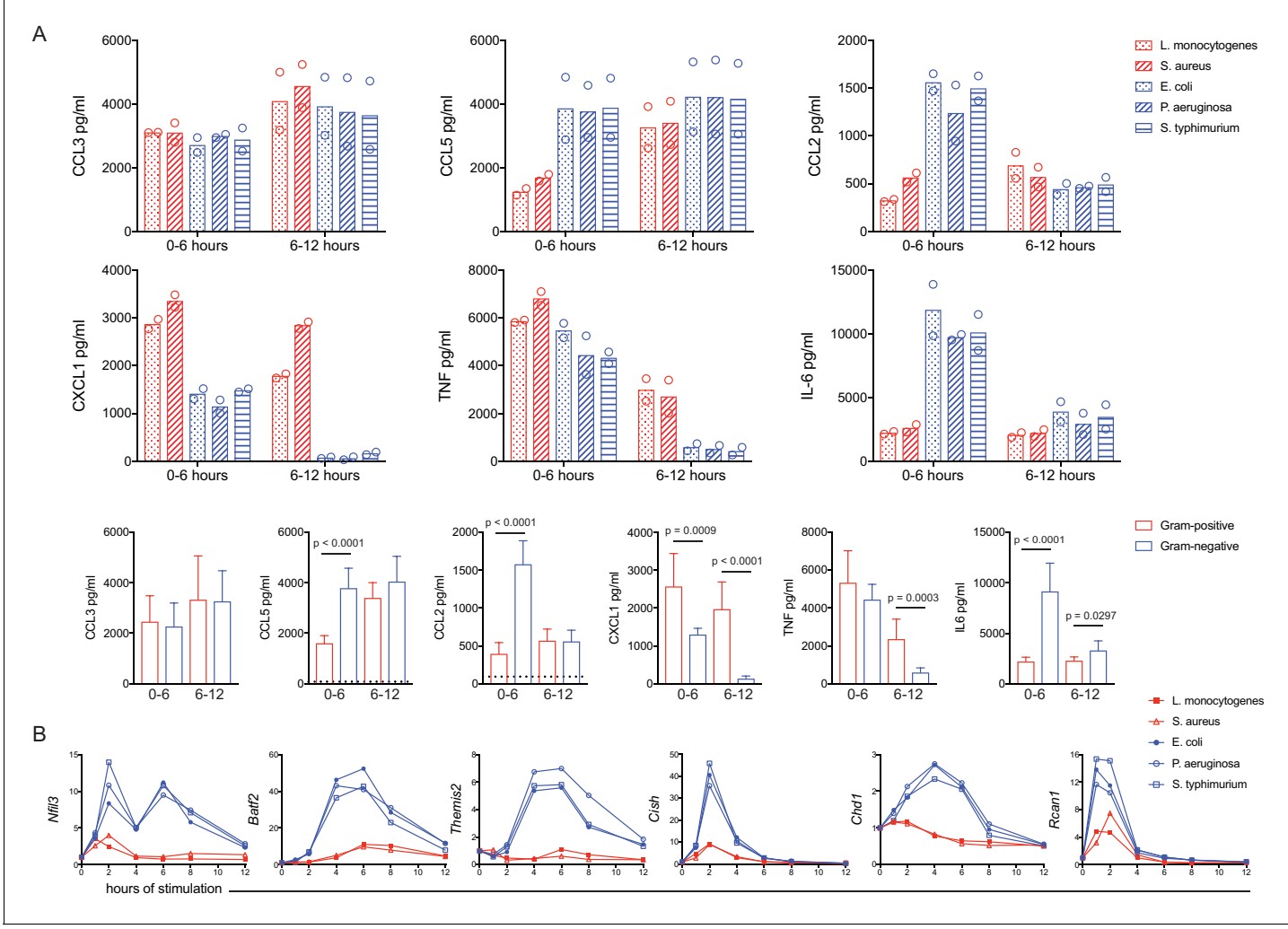

**Figure 3.** A dual TLR titration matrix predicts bacteria-specific cytokine shutdown and regulator induction. BMDM were stimulated with heat-killed bacteria. (**A**) After 6 hr of stimulation, supernatants were removed (0–6 hr) and replaced with fresh medium for a subsequent 6 hr of stimulation (6–12 hr). Cytokines were quantified using cytometric bead array and data points are experimental replicates. Data from bacterial species (upper rows) were combined from three independent experiments (lower row) and statistical significance was determined using unpaired t-tests between Gram-negative- and Gram-positive-stimulated samples; where no p-value is indicated, results were not significant. Dotted lines in the lower row represent the mean values for unstimulated BMDM (0–12 hr). In some cases, these controls are not visible because they are below the level of detection (a zero value). (**B**) Cells were harvested at the indicated time, gene expression was quantified using qPCR, and fold change was calculated compared to unstimulated BMDM. Results are representative of four or more independent experiments.

DOI: https://doi.org/10.7554/eLife.46836.008

The following figure supplements are available for figure 3:

**Figure supplement 1.** Macrophage cytokine secretion in response to varying concentrations of heat-killed bacteria.
DOI: https://doi.org/10.7554/eLife.46836.009

**Figure supplement 2.** Known negative regulators of inflammatory signaling are more highly induced by Gram-negative pathogens.
DOI: https://doi.org/10.7554/eLife.46836.010

is linked to IFN production and this was evident in our microarray data, where TLR4-driven genes included *Ifnb1* and several known IFN-stimulated genes (ISGs), such as *Cxcl10*, *Irf7*, and *Ifit2* (*Figure 2—figure supplement 1*). Thus, we investigated whether shutdown of TNF and CXCL1 was controlled by intracellular signaling events directly downstream of PRR ligation, or dependent on a second, cytokine-mediated signal, possibly involving IFN. For this purpose, cells were stimulated with the heat-killed Gram-negative bacterium *P. aeruginosa*, with and without Brefeldin A (BFA) treatment to prevent cytokine secretion. Consistent with data from supernatant analyses, *Tnf* and

*Cxcl1* mRNA peaked in control-stimulated macrophages at one hour post-stimulation with *P. aeruginosa* and cytokine message decreased between two and six hours (*Figure 4A*). In the presence of BFA, however, this decline did not occur and cytokine mRNA continued to increase through the six hours assessed (*Figure 4A*). In the same samples, we also measured expression of select putative bacteria class-specific regulator genes. In contrast to cytokine expression, which was increased in BFA-treated samples, five of the six regulators showed dramatically decreased induction in the presence of the protein transport inhibitor (*Figure 4B*). These data suggest that the robust regulatory mechanisms induced specifically by Gram-negative bacteria occur downstream of cytokine-mediated feedback, consistent with the temporal delay in the observed bacteria-class specific CXCL1 and TNF shutdown (*Figure 3A*).

To directly examine the possibility that type I IFN production controls the cytokine-mediated feedback we observed, we assessed induction of bacteria class-specific regulator genes in wild type (WT), WT with exogenous IFN-β addition, and *Ifnar1*⁻/⁻ BMDM, which lack the receptor for type I

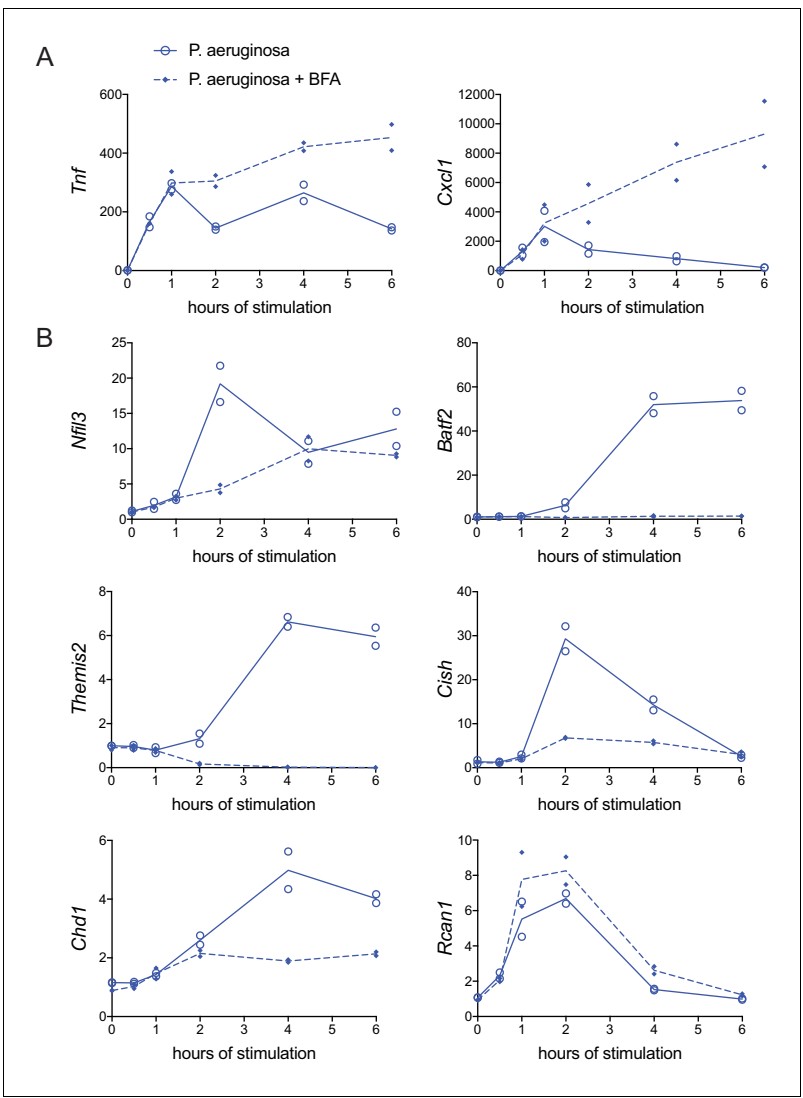

**Figure 4.** Rapid Gram-negative-driven inflammatory response decay and regulator induction are dependent on cytokine feedback. BMDM were stimulated with heat-killed *P. aeruginosa* with or without protein transport inhibitor brefeldin A (BFA). Inflammatory cytokine genes (**A**) and candidate regulator genes (**B**) were quantified by qPCR and fold change was calculated compared to unstimulated BMDM. Data shown are from two independent experiments.
DOI: https://doi.org/10.7554/eLife.46836.011

IFNs. Adding IFN-β to BMDM exposed to the Gram-positive *S. aureus* induced expression of four of the six regulators (*Nfil3*, *Batf2*, *Themis2*, *Chd1*) to levels comparable to Gram-negative bacteria-stimulated BMDM (*Figure 5A*). Consistent with these data, induction of the same four genes was reduced in *Ifnar1⁻/⁻* cells compared to WT upon Gram-negative bacterial stimulation (*Figure 5A*). For the other two candidate regulators, *Rcan1* expression was both cytokine-independent and IFN-independent, whereas *Cish* was cytokine-dependent but IFN-independent (*Figure 4B* and *Figure 5A*).

We next sought to determine if IFN was necessary and sufficient for temporal attenuation of TNF and CXCL1 production, looking for secretion of these cytokines after six hours of stimulation in WT, WT plus IFN-β, and *Ifnar1⁻/⁻* BMDM responding to our panel of heat-killed Gram-positive and Gram-negative bacteria. Indeed, addition of IFN-β resulted in loss of sustained TNF and CXCL1 production in response to stimulation with Gram-positive *L. monocytogenes* and *S. aureus*, showing the rapid loss of secretion characteristic of Gram-negative responses (*Figure 5B*). Conversely, this rapid loss in production was not evident in *Ifnar1⁻/⁻* BMDM, with Gram-negative-stimulated cells continuing to produce TNF and CXCL1 at later time points (*Figure 5B*). Thus, negative feedback by type I IFN supports inflammatory response regulation in a bacteria class-specific manner.

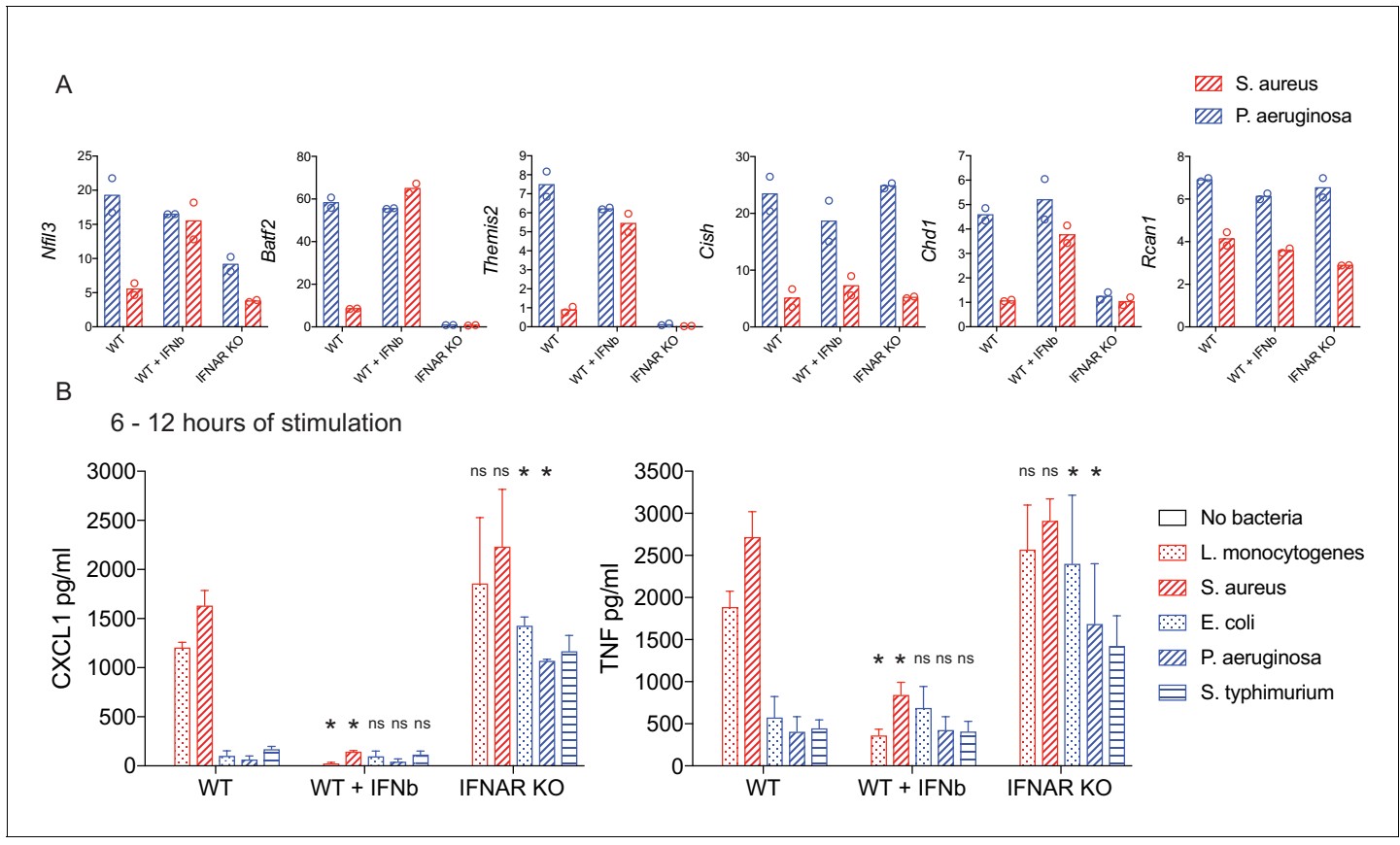

**Figure 5.** IFN-mediated negative feedback supports bacteria class-specific responses. WT or IFNAR KO BMDM were stimulated with heat-killed bacteria, with the addition of IFN-β after one hour of stimulation, where indicated. (**A**) Induced gene expression was measured by qPCR and fold change was calculated compared to unstimulated BMDM. Data are shown for the timepoint of maximum WT expression over a 6 hr time course: 2 hr *Nfil3*, 4 hr *Batf2*, 4 hr *Themis2*, 2 hr *Cish*, 4 hr *Chd1*, and 2 hr *Rcan1*. Data shown are from two independent experiments. (**B**) After 6 hr of stimulation, supernatants were removed and replaced with fresh medium for a subsequent 6 hr of stimulation. The 6–12 hr supernatants were analyzed by cytometric bead array. Cytokine data are pooled from four independent experiments (n = 4, n = 3, and n = 3 for WT, WT + IFN-β, and IFNAR KO, respectively). Stars represent statistical significance based on ordinary two-way ANOVA, comparing WT + IFN-β or IFNAR KO treatment groups to WT for each respective bacterial stimulation (*p≤0.05).
DOI: https://doi.org/10.7554/eLife.46836.012

# IFN regulates inflammatory responses to bacteria independent of IL-10 production

An anti-inflammatory function for type I IFN has been reported in a variety of bacterial infection and inflammatory disease models (*Hamana et al., 2017*; *Katakura et al., 2005*). This activity has been largely attributed to IFN induction of IL-10, which in turn inhibits transcription of TLR-induced inflammatory cytokines including CXCL1, TNF, IL6, and IL1a (*Murray, 2005*). Using our dual TLR ligand titration matrix, we examined whether IL-10 was necessary for bacteria class-specific IFN-mediated negative feedback. We initially noted that there was a strong correlation between *Ifnb1* and *Il10* gene expression in our microarray data (*Figure 6A*). However, we observed that *Il10* expression was influenced by both TLR ligands used in our matrix experiments while *Ifnb1* induction was specifically determined by a TLR4 ligand threshold (*Figure 6A*). Consistent with this pattern of *Ifnb1* expression and autocrine feedback, delayed STAT1 phosphorylation was observed only at high concentrations of TLR4 ligand, while both KLA and P3C concentration influenced acute activation of the NFkB and MAPK pathways (*Figure 6—figure supplement 1*). Disparate dose-related TLR-induced expression of *Ifnb1* and *Il10* suggested that regulation of these mediators may be uncoupled in response to bacterial stimuli (*Figure 6A*). In testing this prediction, we found that *Ifnb1* was robustly induced by the Gram-negative *P. aeruginosa*, consistent with the IFN-dependent cytokine shutdown we observed in response to this species (*Figure 6B*). In contrast, *Il10* expression was more comparable between Gram-positive and Gram-negative stimulation conditions, particularly in the early phase of the response (*Figure 6B*). This early *Il10* induction preceded *Ifnb1* expression and was IFN-independent (*Figure 6B,C*), while sustained *Il10* expression was lost in *Ifnar1*[-/-] BMDM (*Figure 6C*). Consistent with mRNA induction dynamics, we detected STAT3 phosphorylation prior to STAT1 phosphorylation in response to either *S. aureus* or *P. aeruginosa* (*Figure 6—figure supplement 2*).

The uncoupling of type I IFN and IL-10 production led us to consider the possibility that IFN supports contextual regulation of inflammatory control in a manner that is not completely dependent on IL-10. To test this notion, we utilized *Il10rb*[-/-] BMDM, which lack the IL-10 receptor, and assessed expression of the bacteria class-specific candidate genes that were also IFN-regulated. Consistent with previous findings (*Kobayashi et al., 2011*; *Smith et al., 2011*), *Nfil3* was highly dependent on IL-10 for its induction (*Figure 6D*). However, the other IFN-induced genes with negative regulatory function in our assays (*Batf2*, *Themis2*, and *Chd1*) were robustly and selectively upregulated in response to *P. aeruginosa* in both WT and IL-10R KO cells (*Figure 6D*).

Considering that IL-10 is a key regulator of inflammatory cytokine transcription, control of TNF and CXCL1 was perturbed by IL-10R deficiency or IL-10R blockade, as expected (*Figure 6—figure supplement 3*). However, the IFN-dependence of stimulus-specific inflammatory cytokine shutdown was not lost with the addition of exogenous IL-10 (*Figure 6—figure supplement 3*). Thus, we sought to test the model that distinct IFN-mediated and IL-10-mediated negative feedback mechanisms are both required for negative regulation of TNF and CXCL1 (*Figure 6E*, top panel), as opposed to a model whereby the anti-inflammatory properties of IFN are mediated solely though sequential induction of IL-10 (*Figure 6E*, bottom panel). To this end, we stimulated congenically-marked WT and *Ifnar1*[-/-] BMDM separately or in a mixed culture at a 1:1 ratio. In these cultures, we quantified the amounts of CXCL1 produced in response to heat-killed bacteria between 5 and 8 hr of stimulation, when IFN-mediated negative feedback regulation occurs. At this timepoint, a distinct IFN-dependent decline in CXCL1 production was observed in cells stimulated with Gram-negative, but not Gram-positive bacteria (*Figure 6F and G*). The class-specific *Ifnar1-/-* effect, measured as increased CXCL1 production by *Ifnar1*[-/-] BMDM over WT BMDM from Gram-negative stimulation, was seen in both separate and mixed cultures, where WT and KO cells are exposed to comparable levels of IL-10 and other soluble factors produced by the WT cells (IL-10 detected in mixed cultures: *Figure 6—figure supplement 4*). In contrast to CXCL1, at this 5–8 hr timepoint we did not detect the IFN-dependence for TNF regulation that we routinely seen in 6–12 hr supernatant data, and thus we were not able to compare separate and mixed cultures (*Figure 6—figure supplement 4*). The more robust *Ifnar1-/-* effect for CXCL1 at this timepoint, compared to TNF, is consistent with our mRNA time course data showing a more rapid shutdown of *Cxcl1* than *Tnf* (*Figure 2D*). Together, our findings indicate that IFN has critical anti-inflammatory effects independent of IL-10 and other cytokines, through the cell intrinsic action of ISGs.

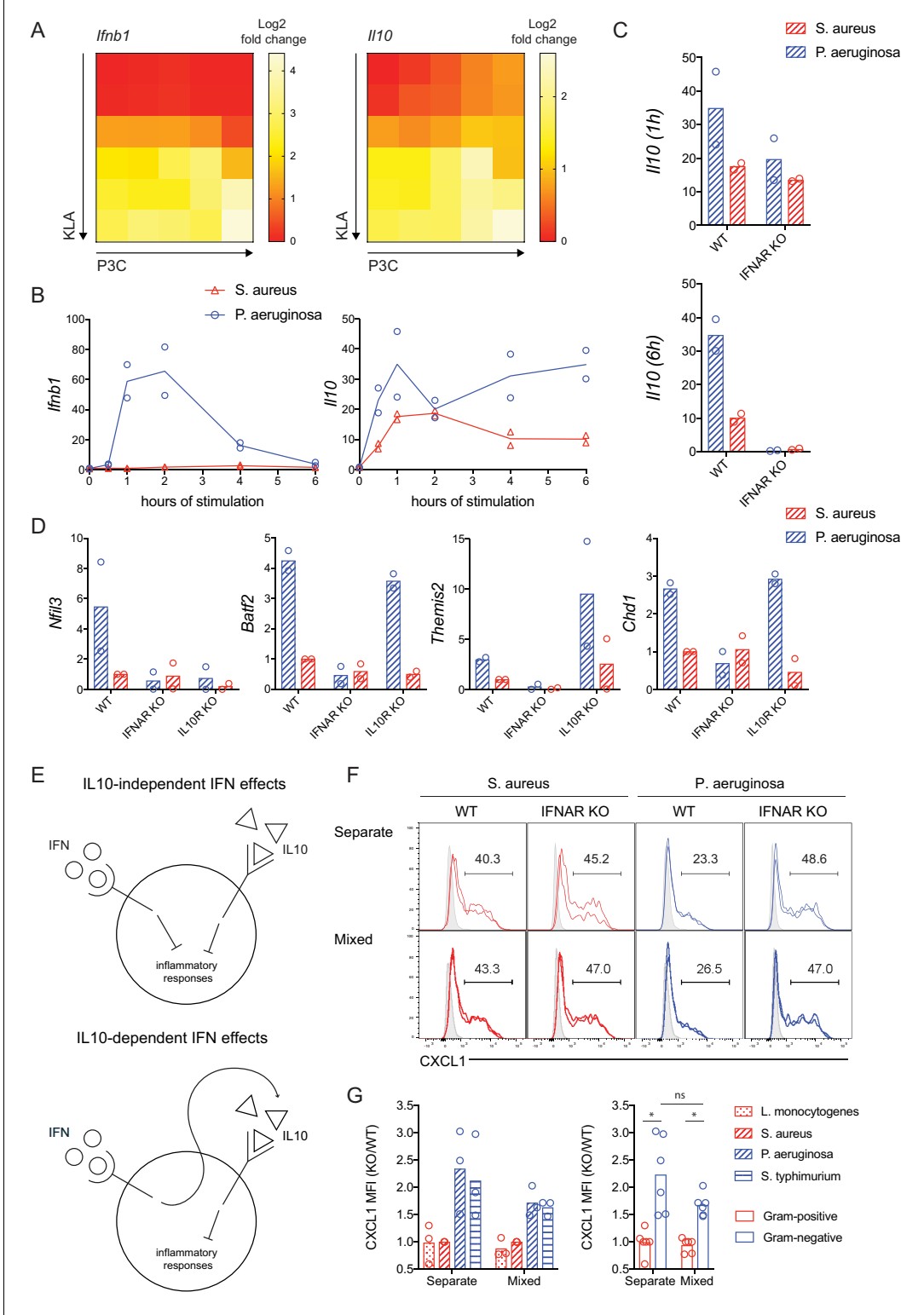

**Figure 6.** IFN regulates inflammatory responses independent of IL-10 production. (**A**) TLR-induced expression of *Ifnb1* and *Il10* at 2 hr, analyzed from microarray data described in *Figure 1*. (**B–D**) WT, IFNAR KO, or IL10R KO BMDM were stimulated with heat-killed *S. aureus* or *P. aeruginosa*, gene expression was measured by qPCR, and fold change was calculated compared to unstimulated BMDM. Data shown represent experimental replicates. In D, the time of maximum expression is plotted (2 hr *Nfil3*, 4 hr *Batf2*, 4 hr *Themis2*, and 4 hr *Chd1*) and the experimental replicates were normalized with respect to WT *S. aureus*. (**E**) Models of IFN and IL10 mediated anti-inflammatory function. (**F, G**). WT (CD45.1+) and IFNAR KO (CD45.1-) BMDM were stimulated with the indicated heat-killed bacteria, separately or in mixed BMDM cultures (1:1 of WT:KO). BFA was added for the final 3 hr of an 8
*Figure 6 continued on next page*

*Figure 6 continued*

hr stimulation. (F) Representative flow cytometry histograms from replicate wells are shown with average frequencies. Shaded histograms represent unstimulated BMDM of the indicated genotype and culture type. (G) CXCL1 quantification as a KO/WT ratio was normalized to WT cells stimulated with *S. aureus* and pooled from three independent experiments. The same data are shown for individual bacterial species or pooled between bacterial class, with stars representing statistical significance based on ordinary one-way ANOVA (*p≤0.05).

DOI: https://doi.org/10.7554/eLife.46836.013

The following figure supplements are available for figure 6:

**Figure supplement 1.** Signaling kinetics of dual-ligand stimulation.
DOI: https://doi.org/10.7554/eLife.46836.014
**Figure supplement 2.** STAT activation upon stimulation with heat-killed bacteria.
DOI: https://doi.org/10.7554/eLife.46836.015
**Figure supplement 3.** The impact of IL10 deficiency or addition on TNF and CXCL1 shutdown.
DOI: https://doi.org/10.7554/eLife.46836.016
**Figure supplement 4.** Characterization of mixed WT and *Ifnar1-/-* BMDM cultures.
DOI: https://doi.org/10.7554/eLife.46836.017

## IFN shapes early inflammatory cytokine dynamics during bacterial lung infection in a pathogen-specific manner

Considering the clinical relevance of understanding mechanisms mediating bacteria class-specific inflammatory cytokine dynamics, we sought to examine our central findings in vivo. *Pseudomonas aeruginosa* and *Staphylococcus aureus* are two of the most common pathogens in hospital acquired pneumonia and these respiratory infections are associated with sepsis and high mortality (*Chastre and Fagon, 2002*; *Mayr et al., 2014*). Thus, there is particular interest in understanding distinct inflammatory control during lung infection with these exemplar Gram-negative and Gram-positive pathogens. While a role for IFN has been described in lung immunity to both of these pathogens (*Cohen and Prince, 2013*; *Martin et al., 2009*), based on our in vitro results we hypothesized that rapid IFN-mediated dampening of CXCL1, TNF, and associated neutrophil recruitment would be specific to Gram-negative *P. aeruginosa*.

We infected *Ifnar1-/-* and WT C57Bl6 littermate controls intranasally with $5 \times 10^7$ CFU of either *P. aeruginosa* and *S. aureus*. Mice were sacrificed at 2- and 6 hr post-infection and bronchoalveolar lavage (BAL) was performed for quantification of cytokine production and neutrophil recruitment. While WT mice infected with *P. aeruginosa* trended towards decreased TNF and CXCL1 from 2 to 6 hr, there was a statistically significant increase in these inflammatory mediators in *Ifnar1-/-* mice at 6 hr, compared to WT (*Figure 7A,B*). CCL2 did not follow the same pattern, consistent with our in vitro findings (*Figure 7C* compared to *Figures 1* and *3*). These results suggest that IFN plays a negative regulatory role in control of select inflammatory mediators at early timepoints after Gram-negative bacterial infection in vivo. We noted a non-significant increase in neutrophil numbers in *P. aeruginosa*-infected *Ifnar1-/-* animals, compared to WT, consistent with the important role of CXCL1 for neutrophil recruitment (*Figure 7D*). Increased TNF and CXCL1 in *P. aeruginosa* infected *Ifnar1-/-* mice was not coincident with changes in CFU, IL-10, or eosinophil recruitment (*Figure 7—figure supplement 1*). While this indicates that sustained early TNF and CXCL1 are not due to increased bacterial burden at this timepoint, there have been reports of increased or decreased bacterial load in *Ifnar1-/-* mice in longer term infection models (*Mancuso et al., 2007*; *O'Connell et al., 2004*; *Perkins et al., 2015*). Thus, we cannot conclude that IFN does not play a role in bacterial clearance. Importantly, we did not see the same IFN-dependent negative feedback in response to *S. aureus* lung infection; CXCL1, TNF, and neutrophil counts were not increased in *S. aureus* infected *Ifnar1-/-* mice compared to WT (*Figure 7*). Together with our in vitro findings, these data support a model whereby distinct triggering of IFN-mediated negative regulatory mechanisms supports bacterial-class specific control of inflammatory cytokine dynamics and the resulting innate cellular response.

## Discussion

Inflammatory cytokines and chemokines, along with the magnitude and kinetics of their production, determine the number, composition, and activity of infiltrating leukocytes in infected sites. Effector mechanisms and response timing must be appropriately regulated to effectively balance pathogen

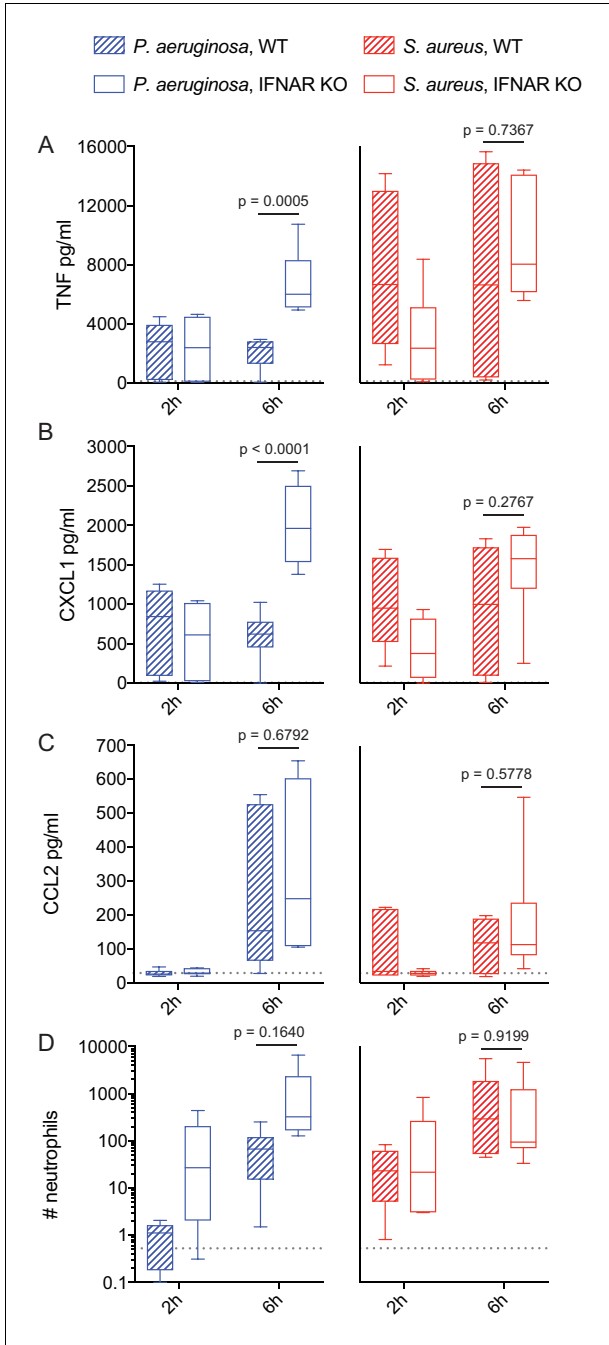

**Figure 7.** IFN-mediated negative feedback shapes in vivo inflammatory responses in a pathogen-specific manner. WT or *Ifnar1-/-* littermate mice were infected intranasally with $5 \times 10^7$ CFU of *P. aeruginosa* or *S. aureus*, as indicated. At 2 and 6 hr after infection, mice were sacrificed and BAL was collected for quantification of cytokines by CBA (A-C) and neutrophils by flow cytometry (D). Data are pooled from two independent experiments, with a total of 6 mice per experimental group. Neutrophil numbers were normalized to the mean of WT *P. aeruginosa* infected mice for each independent experiment. The mean values for PBS treated control mice are shown as dotted lines.

DOI: https://doi.org/10.7554/eLife.46836.018

The following figure supplement is available for figure 7:

**Figure supplement 1.** Lung infection with *P. aeruginosa* and *S. aureus*.

DOI: https://doi.org/10.7554/eLife.46836.019

clearance with the risk of inflammation-associated pathology, and this will vary depending on the nature of the infecting pathogen. Here we have systematically and quantitatively interrogated how macrophages interpret combinatorial PAMP stimuli to develop bacteria class-specific inflammatory responses. Our efforts point to previously unappreciated pathway interactions, whereby distinct stimulus requirements for inflammatory cytokine induction and type I IFN-mediated negative feedback create a broad temporal topology of cytokine and chemokine milieus across dual ligand conditions. These distinct cytokine dynamics were predictive of selective responses to Gram-positive versus Gram-negative bacteria, which were strikingly disparate in their ability to rapidly induce IFN, ISGs, and subsequent inflammatory cytokine attenuation.

Previous reports have found that bacteria-induced gene expression in macrophages was largely recapitulated with soluble TLR4 or TLR2 ligand, for Gram-negative and Gram-positive bacteria respectively (*Nau et al., 2003*). Comparing gene expression downstream of individual TLR ligands often highlights the apparent redundancy in the system, whereby TLR4 responses largely encompass TLR2 responses, with the addition of a TRIF-dependent gene signature. However, non-additive pathway interactions have been described in the context of multi-TLR stimulation, whereby ligand pairs resulted in sustained TLR signaling events, gene expression and cytokine production compared to individual ligands (*Lin et al., 2017*; *Napolitani et al., 2005*; *Tan et al., 2014*). Considering the rapid and robust PAMP-induced production of macrophage-regulating cytokines, cytokine-mediated negative and positive feedback loops can clearly add further layers of non-linear control. Our data demonstrate that distinct quantitative stimulation requirements for TLR synergy and cytokine feedback yield concentration-dependent inflammatory control downstream of TLR engagement. We observed sustained TNF and CXCL1 expression in response to TLR2 ligand in combination with low concentrations of TLR4 ligand, consistent with a greater cumulative signaling in the presence of both stimuli. However, above a threshold of TLR4 exposure we observed a rapid decay in production of these cytokines, despite robust and sustained NFkB and MAPK activation. This loss of expression was mediated by a negative feedback loop dependent on signaling in response to type I IFN, with IFN expression being dictated by the magnitude of TLR4 ligation, largely independent of TLR2 exposure levels. Thus, select PAMP conditions can yield positive effects of combinatorial multi-ligand stimulation while avoiding negative regulatory mechanisms associated with strong stimulation through one particular receptor pathway.

Though typically associated with viral infections, type I IFNs are also robustly induced by bacteria. This can occur through various PRRs, all of which signal after the bacteria has been internalized into membrane-bound compartments by the host cell (*Boxx and Cheng, 2016*). In recent years, it has become clear that LPS induces a strong IFN response through its interaction with TLR4 in bacteria-containing endosomes, where TLR4 preferentially signals through the TRIF-TRAM pathway (*Kagan et al., 2008*; *Zanoni et al., 2011*). Consistent with these observations, previous reports have shown a strong IFN response to heat-killed Gram-negative bacteria (*Sing et al., 2000*). Stimulation of macrophages in vitro with heat-killed bacteria in the current study was motivated by our efforts to reduce variables that can be associated with live bacteria, including a plethora of virulence factors, as well as replication rates and bacterial motility. Recognition of PAMPs associated with living microbes (or 'vita-PAMPs') have been linked to IFN production and inflammasome activation in both Gram-negative and Gram-positive bacteria (*Sander et al., 2011*). However, while LPS from Gram-negative bacterial cell walls is likely one of the first PAMPs sensed, vita-PAMPs need to gain access to the cytosol before activating sensors such as STING (*Moretti et al., 2017*). Thus, we speculate that IFN-mediated negative feedback will occur in response to some live Gram-positive pathogens, but with different timing and consequences compared to Gram-negative bacteria. This hypothesis is consistent with our in vivo findings, whereby IFN-dependent negative regulation of TNF and CXCL1 was observed within 6 hr of *P. aeruginosa* lung infection, but not in response to *S. aureus*.

Once type I IFNs are produced, these cytokines act in a paracrine and autocrine fashion, binding to the type I IFN receptor (IFNAR). Binding of IFN to IFNAR leads to the activation of transcription factors and complexes, including STAT1 homodimers and IFN-stimulated gene factor 3 (ISGF3), made up of STAT1, STAT2, and IRF9 (*Schreiber, 2017*). These factors contribute to the transcription of hundreds to thousands of ISGs, the majority of which have unknown function in the context of bacterial immunity (*Schneider et al., 2014*). The described functions of type I IFNs in bacterial infection are diverse, including anti-microbial and pro-inflammatory roles. The majority of LPS secondary response genes require type I IFN signaling for their sustained expression (*Tong et al., 2016*), and

type I IFN signaling has been recently reported to enhance NFκB activity (*Mitchell et al., 2019*). Many ISGs are potent inflammatory chemokines, such as CXCL10, a chemotactic factor for monocytes vital for attracting these phagocytes to areas of infection or damage (*Petrovic-Djergovic et al., 2015*). Additionally, the guanylate-binding proteins (GBPs) - small, IFN-regulated, cellular proteins found in macrophages and other cells - target bacterial membranes, lysing the bacteria in the process (*Liu et al., 2018*; *Wallet et al., 2017*). In contrast, IFNs are often associated with anti-inflammatory action (*Boxx and Cheng, 2016*; *Hamana et al., 2017*; *Katakura et al., 2005*). Consistent with this, the TRIF pathway can interfere with MyD88 pathway-driven inflammatory gene expression (*Amit et al., 2009*). However, the anti-inflammatory capacity of ISGs has not been thoroughly studied, with the exception of IFN-induced IL-10 (*Iyer et al., 2010*; *McNab et al., 2014*), which itself orchestrates a plethora of anti-inflammatory functions (*Rojas et al., 2017*).

In this study, we demonstrate ISG induction and associated shutdown of inflammatory cytokine production independent of IL-10. *Batf2*, *Themis2*, and *Chd1* were induced by Gram-negative stimuli in a highly IFN-dependent manner, and siRNA knockdown studies revealed the capacity of the corresponding gene products for negative regulatory function. Together with the results of WT and *Ifnar1*$^{-/-}$ mixed culture studies, these data indicate that IFN has direct effects on the inflammatory potential of macrophages via cell-intrinsic ISG mechanisms. Our results suggest that this negative regulation will be invoked in a bacteria class-specific manner, with early IFN-mediated feedback and downstream ISG expression being selectively triggered by Gram-negative bacteria through their expression of LPS.

Given evidence that IL-10, a well-described anti-inflammatory cytokine, can be induced downstream of IFN signaling (*Chang et al., 2007*; *McNab et al., 2014*), our findings suggest that IL-10-dependent and independent IFN-mediated mechanisms may play non-redundant anti-inflammatory roles in bacterial infection. In our experimental system, blocking IL-10 signaling abrogated negative regulation of inflammatory signals, despite not affecting the transcription of IFN-dependent negative regulators. However, this effect was seen regardless of the class of bacterium sensed. We also noted that IFNAR deficiency impacted late IL-10 induction, but not at one hour of stimulation, and that late IL-10 was IFN-dependent in response to both Gram-negative and Gram-positive bacteria. We speculate that rapid IL-10-independent IFN-mediated feedback mechanisms are initiated by surface TLR4 ligation. This may happen in response to live bacteria, killed bacteria, or soluble LPS, but in a manner that is specific to Gram-negative pathogens. In contrast, later IFN-driven IL-10-dependent feedback mechanisms may be triggered in response to either Gram-negative or Gram-positive bacteria, likely enhanced by bacterial internalization and ligation of endosomal or cytosolic PRRs. Thus, IL-10-independent and dependent IFN feedback mechanisms are likely temporally distinct and, depending on the timing and magnitude of IFN induction, support fine tuning of pathogen-specific inflammatory cytokine dynamics. Further studies will be needed to experimentally test the relative contributions of IFN-mediated IL-10-dependent and independent negative regulation in shaping of early inflammatory responses to difference types of bacterial pathogens.

The class of bacterial pathogen impacts monocyte and macrophage production of inflammatory cytokines and, in cases of human bacteremia, Gram-negative organisms are generally associated with more robust inflammatory responses and poorer clinical outcomes during sepsis and septic shock, as compared to Gram-positive bacteremia (*Abe et al., 2010*; *Hessle et al., 2005*; *Surbatovic et al., 2015*). We propose that induction of IFN through the TLR4-TRIF pathway evolved as a buffer of inflammation, preventing an overreaction to LPS. IFNAR signaling has dramatic effects on outcome during infection with both Gram-negative and Gram-positive species (reviewed in *Boxx and Cheng, 2016*). Several studies found type I IFN to be beneficial during bacterial infection, promoting tissue integrity and suppressing excessive inflammatory responses. However, there are also examples of these anti-inflammatory effects being detrimental to the host, dampening host-protective responses, as in the case of *Mycobacteria tuberculosis* and post-influenza bacterial pneumonia (*Mayer-Barber et al., 2014*; *Nakamura et al., 2011*). In trying to understand response variation across bacterial infections, our in vitro and in vivo data highlight the importance of comparing inflammatory response dynamics, in addition to magnitude. Future studies may reveal a trend towards prolonged and damaging inflammatory responses to Gram-positive infection, particularly with respect to CXCL1-dependent neutrophil recruitment, as a result of diminished or delayed IFN-mediated feedback.

Therapeutic trials targeting mediators of the inflammatory response have also reported distinct responses for sepsis patients with Gram-positive vs. Gram-negative infections (*Gao et al., 2008*). While more comprehensive efforts have failed to find an association between pathogen class or species and the outcome of severe sepsis and septic shock (*Zahar et al., 2011*), systematic understanding of how inflammation differs based on the type of bacterial pathogen is of considerable clinical relevance. Elucidating the molecular mechanisms that support distinct responses could yield insight into tailored therapies. Furthermore, in-depth analyses of ISGs and their roles in antibacterial responses will provide us with a better understanding of the differences between these bacteria and how the immune system has evolved inflammatory responses tailored to combat them.

# Materials and methods

## Key resources table

| Reagent type (species) or resource | Designation | Source or reference | Identifiers | Additional information |
|---|---|---|---|---|
| Biological sample | Kdo2-Lipid A Di[3-deoxy-D-manno- octulosonyl]-lipid A (ammonium salt) | Avanti Polar Lipids, Inc | 699500 | |
| Biological sample | Pam3CSK4 Synthetic triacylated lipopeptide | InvivoGen | Tlrl-pms | |
| Sequence-based reagent | Ambion Silencer Select siRNA library | ThermoFisher Scientific | AM81810 | |
| Commercial assay, kit | Lipofectamine RNAiMAX transfection reagent | ThermoFisher Scientific | 13778030 | |
| Strain, strain background (*Pseudomonas aeruginosa*) | PA01 | ATCC | BAA-47 | |
| Strain, strain background (*Salmonella enterica enterica* serovar Typhimurium) | 14028 | ATCC | 14028 | |
| Strain, strain background (*E. coli*) | K12 MG1655 | ATCC | 700926 | |
| Strain, strain background (*Stapylococcus aureus*) | FDA209 | ATCC | 6538P | |
| Strain, strain background (*Listeria monocytogenes ΔactA*) | DPL1942 | (*Brundage et al., 1993*) | | Deletion mutant for gene *actA* |
| Strain, strain background (*Mus musculus*) | C57BL/6J | Jackson Laboratory | 000664 | |
| Strain, strain background (*Mus musculus*) | B6.129S2-Ifnar1$^{tm1Agt}$/Mmjax | Jackson Laboratory | 32045-JAX | Knockout mutant for gene *Ifnar1* |
| Strain, strain background (*Mus musculus*) | B6.129S2-Il10rb$^{tm1Agt}$/J | Jackson Laboratory | 005027 | Knockout mutant for gene *Il10rb* |
| Antibody | Rat anti-mouse IL10R monoclonal | Biolegend | 112709 | Blocked with 10 µg/mL |
| Recombinant protein | Mouse IL-10 | Peprotech | 210–10 | |

## Mouse macrophages and cell culture

Mice were maintained in specific-pathogen-free conditions and all procedures were approved by the NIAID Animal Care and Use Committee (National Institutes of Health, Bethesda, MD). Bone marrow progenitors isolated from sex-matched wild-type (strain 000664), *Ifnar1-/-* (strain 32045), and *Il10rb-/-* (strain 005027) mice on a C57BL/6J background (Jackson Laboratories) were differentiated into BMDM during a 6 day culture in complete Dulbecco's modified Eagle's medium (DMEM + 10% FBS, 100 U/ml penicillin, 100 U/ml streptomycin, 2 mM L-Glutamine, 20 mM HEPES) supplemented with 60 ng/ml recombinant mouse M-CSF (R and D systems). One day prior to stimulation, cells were rinsed with cold PBS, then scraped from plates using a cell lifter. Cells were then plated in the appropriate tissue-culture-treated plate in complete DMEM and allowed to rest overnight at 37° C, 5% CO2, 95% relative humidity prior to stimulation.

## Bacteria and purified ligands

Bacterial strains used were as follows: *Pseudomonas aeruginosa* (PA01), *Salmonella enterica Typhimurium* (14028 s), *Escherichia coli* (K12 MG1655), *Staphylococcus aureus* (FDA209), and *Listeria monocytogenes* (ΔactA strain DPL1942; Brundage et al., 1993). Bacteria were grown up overnight in either Luria-Bertani Miller broth (*Pseudomonas, Salmonella,* and *E. coli*) or trypticase soy broth (*Staphylococcus* and *Listeria*) shaking at 37° C. Absorbance at 600 nm was measured to determine bacterial concentration. Bacteria were washed 2x in PBS, then heat-killed at 60–95° C (depending on bacteria) for 1 hr. Heat-killed bacteria were stored at 4° C. Bacteria were diluted to $1 \times 10^9$ CFU/mL in PBS, representing a 10X solution (1X being equivalent to MOI of 100), and Kdo2-LipidA (KLA, Avanti Polar Lipids) and Pam3CSK4 (P3C, Invivogen) were diluted in complete DMEM, before addition to BMDM.

## BMDM stimulation

BMDM were plated at $1 \times 10^6$ cells/mL in 96-well plates (100 μL/well) for supernatant analyses, or 48-well plates (200 μL/well) for flow cytometry or RNA analyses. 16–20 hr after plating, 10X solutions containing heat-killed bacteria or purified TLR ligand(s) were added. Control wells received 10% (by volume of media in the well) of PBS (Gibco) or complete DMEM. For intracellular cytokine staining, Brefeldin A (BFA, BD Golgi plug) was added to cells 3 hr prior to fixation. When used to prevent cytokine feedback, BFA was added to cells 30 min post-stimulation. Cells used for RNA analysis were lysed in 200 μL of TriReagent (Zymo Research) and plates were then frozen at −80° C until RNA extraction was performed. For cytokine analyses, supernatants were removed from stimulated cells and stored at −80° C until analysis. When comparing early and late responses, 'early' supernatants were removed at 6 hr post-stimulation and replaced with fresh warm complete DMEM. Cells were then put back in the incubator for a further 6 hr and 'late' supernatant was harvested. For IL-10 addition experiments, recombinant IL-10 (1 ng/ml; Peprotech) was added to cells 2 hr after the start of stimulation and then added again when supernatants were replaced for the final 6 hr. In appropriate experiments, IL10R blocking antibody (10 μg /ml; Biolegend) was added 30 min prior to BMDM stimulation with heat-killed bacteria.

## HEK cell culture and stimulation

TLR2- and TLR4-expressing HEK293 cells were purchased from Invivogen. Cells were cultured in complete DMEM with of 10 μg/mL blasticidin (Invivogen) to ensure TLR expression. HEK293-TLR cells were plated and stimulated with heat-killed bacteria, as described for BMDM. After 20 hr of stimulation, supernatants were collected and IL-8 was quantified by ELISA (DuoSet; R and D Systems) according to the manufacturer's directions.

## Cytometric bead array

Cytometric bead array (CBA) was performed using the mouse/rat soluble protein master buffer kit combined with the appropriate CBA flex sets (BD Biosciences), according to manufacturer's instructions. Supernatants were diluted 2x in assay diluent before mixing with capture beads. Flow data were collected on a BD Fortessa and analyzed with FlowJo. Data represent the median fluorescence intensity of PE on beads collected for analysis, extrapolated to protein concentration using a standard curve.

## Western blot

For Western blotting, cells were lysed on ice (10 mM TrisHCl, 140 mM NaCl, 2 nM EDTA, 1% NP40 lysis buffer containing Roche PhosSTOP and cOmplete ULTRA phosphatase and protease inhibitors) and boiled with SDS reducing sample buffer. Samples were electrophoresed on 4–20% Novex Tris-Glycine Gels (Invitrogen), transferred to nitrocellulose (Bio-Rad), and probed with p-STAT1 (Y701) (Cell Signaling Technology), p-STAT3 (Y705) (Cell Signaling Technology), or pp-p38 (T180/Y182) (Cell Signaling Technology) antibodies, followed by HRP-conjugated secondary antibodies. Blots were developed with SuperSignal Chemiluminescence Substrate (Thermo Scientific).

## Microarray

RNA extraction was performed using RNeasy Mini Kit (Qiagen). RNA samples were amplified and labeled using the Illumina TotalPrep RNA Amplification kit (Applied Biosystems) with an input of 500 nanograms of total RNA per sample. Biotinylated aRNA was hybridized to Illumina MouseWG-6 v2.0 expression beadchip (NCBI/GEO Accession GPL6887) and imaged using the Illumina HiScan-SQ, following standard Illumina protocols. Signal data was extracted from the image files with the Gene Expression module (v. 1.9.0) of the GenomeStudio software (v. 2011.1) from Illumina, Inc, and signal intensities were converted to log2 scale. Data for array probes with insufficient signal (detection p-value<0.1 in at least two arrays) were considered 'not detected' and were removed from the dataset before normalization. Quantile normalization was applied using JMP/Genomics software version 6.0 (SAS Institute Inc, Cary NC). GEO accession number GSE117842.

## siRNA gene knock-down

BMDM were transfected with siRNA (Ambion Silencer Select) against regulator candidates using Lipofectamine RNAiMAX (Invitrogen). 2.5 ul of siRNA (0.8 uM) was spotted into 384 well plates and 0.4 ul of RNAiMAX premixed with 5.1 ul of serum-free complete DMEM (DMEM, 2 mM L-Glutamine, 20 mM HEPES) was added to each well. Plates were tapped to mix the transfection reagents and centrifuged at 400 g for 1 s to bring solution to the bottom of the wells. After incubation for 30 min at room temperature, 20,000 BMDMs in 32 µl of complete 1.25X FBS DMEM (DMEM + 12.5% FBS, 100 U/ml penicillin, 100 U/ml streptomycin, 2 mM L-Glutamine, 20 mM HEPES) were seeded per well for a final siRNA concentration of 50 nM. Plates were incubated at 37°C in a humidified atmosphere with 5% $CO_2$ for 48 hr. Cells were stimulated by replacing medium with 50 ul fresh medium containing the indicated TLR ligand. After 6 hr, supernatant was collected and viability was assessed using CellTiter-Glo (Promega). Secreted cytokine was quantified in supernatants by ELISA (DuoSet; R and D Systems) and values were normalized based on cell viability. In cases where siRNA treatment resulted in less than 50% of control viability, siRNA were excluded from the analysis.

## RNA isolation, microfluidic qPCR, and real-time qPCR

RNA isolation was performed using the Direct-ZOL-96 RNA extraction kit (Zymo Research), according to manufacturer's directions. RNA was quantified using a Nanodrop. With the exception of *Figure 6*, gene expression was quantified by microfluidic qPCR, as previously described (*Gottschalk et al., 2016*). In brief, RNA was reverse transcribed using Superscript VILO cDNA synthesis system (Invitrogen); cDNA was combined with pooled primer sets (Deltagene assays, Fluidigm) and Taqman Preamp Master Mix (Applied Biosystems). Specific Target Amplification was run on a CFX Connect thermal cycler (Bio-rad). Following pre-amplification, unincorporated primers were digested by the addition of Exonuclease 1 (New England Biolabs). Samples were analyzed by qPCR on the Fluidigm Biomark instrument using 96.96 chips according to manufacturer's instructions (Fluidigm). Data were exported from Fluidigm Real-time PCR Analysis software version 3.1.3, using the Linear (Derivative) Baseline method, a global threshold of 0.01, and a 0.65 quality threshold, parameters that were found to exclude non-specific amplification and reduce plate-plate variation. For data in *Figure 6* and *Figure 6—figure supplement 1*, RNA was reverse transcribed using a cDNA synthesis kit (BioRad), according to manufacturer's instructions. RT-PCR was performed using TaqMan probes (Thermo Scientific) on the ABI Quantstudio 6. Analyses were performed using the ΔΔCt method comparing genes of interest to an *Actb* control.

## Flow cytometry

For intracellular cytokine staining, cells were fixed at the end of stimulation by addition of paraformaldehyde to cell cultures at a final concentration of 1.6%, followed by a 5 min incubation at room temperature (RT). After two washes with PBS + 1% FBS, cells were gently scraped from plates for staining. Intracellular cytokine samples were processed using BD Cytofix/Cytoperm reagents, as directed, blocked using Fc receptor specific antibody (24G2, eBioscience), and stained with anti-TNF (eBioscience, MP6-XT22) or anti-CXCL1 (R and D, 1174A) antibodies. Signaling samples were permeabilized using ice cold MeOH for 1–18 hr at −20C, blocked using 5% goat serum and Fc receptor specific antibody, and stained for 1 hr at RT with anti-IκBα (Cell Signaling Technology, L35A5), anti-Erk (phospho-Thr202/Tyr204, Cell Signaling Technology, D13.14.4E or 197G2), anti-STAT1 (phospho-Tyr701, Cell Signaling Technology, 58D6) and/or anti- p38 (phospho-Thr180/Try182, BD Biosciences, 36/p38). Data were collected on a BD Fortessa and analyzed in FlowJo.

## Linear regression analysis

Linear regression models were developed to identify the transcriptional response determined by TLR2 and TLR4 ligand concentration, individually or in combination. To filter out genes showing no variance across different stimulations, Median Absolute Deviation (MAD) for each gene was calculated and genes with 0 MAD were filtered out. For each gene, four different models (shown in *Figure 1D*) were developed to fit the microarray transcriptional data across dual TLR ligand matrix stimulation conditions. These models had different levels of complexity and captured the effect of TLR2 and TLR4 stimulations individually or with their additive and interactive effects. As the number of predictors increased with model complexity, $R^2$ values were adjusted for the number of predictors in the model. For calculating MAD and $R^2$ values from linear model, `mad()` and `lm()` functions were used from the `{stats}` package of the base R. Adjusted $R^2$ values of model were obtained from the model using the `summary()` function.

## Lung infection

WT and *Ifnar1*-/- mice were generated by breeding *Ifnar1*+/- mice together and genotyping pups at the time of weaning using the REDExtract-N-Amp Tissue PCR Kit (Sigma) according to manufacturer's instructions. Genotyping PCRs were performed using the following primers: (Forward) CGAGGCGAAGTGGTTAAAAG, (WT-Reverse) ACGGATCAACCTCATTCCAC, and (KO-Reverse) AATTCGCCAATGACAAGACG. When mice reached 8 to 12 weeks of age, sex-matched cohorts were anaesthetized with isoflurane before being inoculated intra-nasally with 50 µL of either *P. aeruginosa* strain PA01 or *S. aureus* strain FDA309 diluted to $1 \times 10^9$ CFU/mL in PBS. Control mice were given 50 µL of PBS. Mice were allowed to wake up and were placed in boxes with food and water for 2 or 6 hr. At these time points, mice were euthanized by $CO_2$ inhalation. Broncho-alveolar lavage (BAL) was performed by flushing the lungs three times using 1 mL of ice-cold PBS + 2 mM EDTA, followed by the removal of the right lung, which was stored in PBS at 4˚C overnight. BALs were centrifuged at 500 g for 5 min to separate cells and supernatant. Cells were then stained for flow cytometry (see above) while protein levels in supernatants were quantified by CBA assay (see above). CFUs were quantified by mashing the right lungs in 1 mL of PBS before performing serial dilutions and plating on trypticase soy agar plates supplemented with 5% defibrinated sheep's blood (Remel) and incubating overnight at 37˚C.

## Acknowledgements

We thank A Nita-Lazar, SJ Vayttaden, B Lin and members of the RNG laboratory and the Laboratory of Immune System Biology for advice and helpful discussions, A Gola for assistance with experiments, and MK Atianand and N Subramanian for critical reading of the manuscript. We also acknowledge the Genomic Technologies Section, Research Technologies Branch, National Institute of Allergy and Infectious Diseases, including F Otaizo-Carrasquero for microarray hybridizations and TG Myers for microarray data processing. This work was supported by the Intramural Research Program of the National Institute of Allergy and Infectious Diseases, NIH.

## Additional information

### Competing interests
Ronald N Germain: Reviewing Editor, eLife. The other authors declare that no competing interests exist.

### Funding

| Funder | Grant reference number | Author |
| --- | --- | --- |
| National Institute of Allergy and Infectious Diseases | Intramural Research Program | Ronald N Germain |

The funders had no role in study design, data collection and interpretation, or the decision to submit the work for publication.

### Author contributions
Rachel A Gottschalk, Conceptualization, Formal analysis, Investigation, Visualization, Methodology, Writing—original draft; Michael G Dorrington, Formal analysis, Investigation, Writing—original draft; Bhaskar Dutta, Formal analysis, Visualization, Writing—review and editing; Kathleen S Krauss, Stefan Uderhardt, Waipan Chan, Investigation; Andrew J Martins, Formal analysis, Investigation, Methodology; John S Tsang, Iain DC Fraser, Supervision, Writing—review and editing; Parizad Torabi-Parizi, Investigation, Methodology, Writing—review and editing; Ronald N Germain, Conceptualization, Supervision, Writing—review and editing

### Author ORCIDs
Rachel A Gottschalk (iD) https://orcid.org/0000-0002-5248-8816
Ronald N Germain (iD) https://orcid.org/0000-0003-1495-9143

### Ethics
Animal experimentation: Mice were maintained in specific-pathogen-free conditions and all procedures were approved by the NIAID Animal Care and Use Committee (National Institutes of Health, Bethesda, MD) under protocol LISB-4E.

### Decision letter and Author response
Decision letter https://doi.org/10.7554/eLife.46836.024
Author response https://doi.org/10.7554/eLife.46836.025

## Additional files

### Supplementary files
• Transparent reporting form
DOI: https://doi.org/10.7554/eLife.46836.020

### Data availability
Microarray data have been deposited in GEO under accession number GSE117842.

The following dataset was generated:

| Author(s) | Year | Dataset title | Dataset URL | Database and Identifier |
| --- | --- | --- | --- | --- |
| Gottschalk RA, Fraser ID, Germain RN | 2019 | BMDM dual TLR stimulation | http://www.ncbi.nlm.nih.gov/geo/query/acc.cgi?acc=GSE117842 | NCBI Gene Expression Omnibus, GSE117842 |

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
