## [Decision Letter]

Thank you for submitting your article "IFN-mediated negative feedback supports bacteria class-specific macrophage inflammatory responses" for consideration by *eLife*. Your article has been reviewed by three peer reviewers, and the evaluation has been overseen by a Reviewing Editor and Wendy Garrett as the Senior Editor. The following individuals involved in review of your submission have agreed to reveal their identity: Alexandre Corthay (Reviewer #1); Jakob Zimmermann (Reviewer #3).

The reviewers have discussed the reviews with one another and the Reviewing Editor has drafted this decision to help you prepare a revised submission.

Summary:

We appreciated your work showing how macrophages respond differently to Gram-positive and Gram-negative bacteria in order to facilitate a pathogen-tailored protective innate immune response. The main findings were that i) with combinations of two TLR agonists the strongest cytokine response could be obtained with concentrations of TLR ligands that would be suboptimal when used alone; and ii) Gram-negative bacteria induce a negative feedback loop that is type I interferon (IFN) dependent with a downregulatory effect on the production of pro-inflammatory cytokines by macrophages. These are important findings.

Essential revisions:

In consultation with the reviewers, we are asking for critical revisions that address three areas.

1) The rationale for use of P3C in discrimination of the bacterial responses and better insight into the actual contributions of TLR2 and TLR4 for the different bacteria used.

The reason for selecting KLA was considered logical. As mentioned in the second paragraph of the subsection “TLR pathway non-additivity supports context-specific inflammatory responses”, KLA is the stimulatory unit of LPS that is found exclusively on Gram-negative bacteria. In contrast, the rationale for selecting P3C needs to be better explained. P3C is a synthetic bacterial lipoprotein that binds to TLR-2 (Aliprantis et al., 1999). Bacterial lipoproteins are produced by both Gram-positive and Gram-negative bacteria. Therefore, TLR-2-mediated recognition of bacterial lipoproteins cannot be used by macrophages to selectively recognize Gram-positive bacteria in a similar manner as TLR-4 detection of LPS allows specific recognition of Gram-negative bacteria. These facts should be made clear in the manuscript and the authors need to explain which model they propose.

Data should be provided to show whether the five selected bacterial strains (*L. monocytogenes, S. aureus, E. coli, P. aeruginosa, S. typhimurium*) express ligands (agonists) for TLR-2 and TLR-4. This could be done using reporter cell lines expressing TLR-2 and TLR-4.

2) The issue of verifying that IFN-β effects are independent of IL-10.

While the possibility for ISGs to negatively regulate the inflammatory response independent of IL-10 was convincing, additional experiments are considered necessary to confirm the extent to which IFN-β is indeed independent of IL-10 in the contexts presented:

a) In the mixed co-culture assay in Figure 6F-G, the authors state that IL-10 is present but it would be good to directly measure it (via RT-PCR or ELISA) to confirm that the level is comparable. In addition, blocking IL-10R with a blocking antibody in these mixed cultures would confirm the extent to which this effect is IL-10 independent. (Minor note: the details of the ratios of the mixed culture experiment in the Materials and methods were not apparent.)

b) The results in Figure 2C-D, and Figure 6D and 6F-G are interpreted together to establish the negative regulation of IFN-β independent of IL-10. However, TNF is the readout in Figure 2 and CXCL1 in Figure 6F-G. Since TNF and CXCL1 are observed to show similar "hot spot" behavior under combinations of gram positive and negative bacteria, it would be reasonable to test both of these, particularly in Figure 6F-G.

3) Demonstration of the effects in unstimulated controls.

A few experiments/figures would benefit from an untreated control group (usually BMDM that have not been exposed to heat-killed bacteria) to be able to fully interpret the data. This concerns Figures 1A, 3A, 5A, 5B, 6F and Figure 6—figure supplement 3.

---

## [Author Response]

Essential revisions:In consultation with the reviewers, we are asking for critical revisions that address three areas.1) The rationale for use of P3C in discrimination of the bacterial responses and better insight into the actual contributions of TLR2 and TLR4 for the different bacteria used.The reason for selecting KLA was considered logical. As mentioned in the second paragraph of the subsection “TLR pathway non-additivity supports context-specific inflammatory responses”, KLA is the stimulatory unit of LPS that is found exclusively on Gram-negative bacteria. In contrast, the rationale for selecting P3C needs to be better explained. P3C is a synthetic bacterial lipoprotein that binds to TLR-2 (Aliprantis et al., 1999). Bacterial lipoproteins are produced by both Gram-positive and Gram-negative bacteria. Therefore, TLR-2-mediated recognition of bacterial lipoproteins cannot be used by macrophages to selectively recognize Gram-positive bacteria in a similar manner as TLR-4 detection of LPS allows specific recognition of Gram-negative bacteria. These facts should be made clear in the manuscript and the authors need to explain which model they propose.Data should be provided to show whether the five selected bacterial strains (L. monocytogenes, S. aureus, E. coli, P. aeruginosa, S. typhimurium) express ligands (agonists) for TLR-2 and TLR-4. This could be done using reporter cell lines expressing TLR-2 and TLR-4.

To our knowledge, there is no PRR that is ligated exclusively by Gram-positive bacteria. We expected both Gram-negative and Gram-positive bacterial products to ligate TLR2 and chose P3C because it is a pure and highly quantifiable TLR2 ligand. We modified the text to clarify this, including discussion of new data we have added using TLR2- and TLR4-expressing HEK293 cells to show that all 5 selected bacterial strains ligate TLR2, while TLR4 ligation is specific to the three Gram-negative species (Figure 1—figure supplement 1).

2) The issue of verifying that IFN-β effects are independent of IL-10.While the possibility for ISGs to negatively regulate the inflammatory response independent of IL-10 was convincing, additional experiments are considered necessary to confirm the extent to which IFN-β is indeed independent of IL-10 in the contexts presented:a) In the mixed co-culture assay in Figure 6F-G, the authors state that IL-10 is present but it would be good to directly measure it (via RT-PCR or ELISA) to confirm that the level is comparable. In addition, blocking IL-10R with a blocking antibody in these mixed cultures would confirm the extent to which this effect is IL-10 independent. (Minor note: the details of the ratios of the mixed culture experiment in the Methods were not apparent.)

The mixed WT/IFNAR KO co-culture system (text noting 1:1 ratio has been added) allows us to detect IFN-dependent mechanisms in conditions where cells of both genotypes are exposed to comparable levels of IL-10 and other soluble factors produced by the WT cells. We now include data in Figure 6—figure supplement 4 reporting IL-10 levels in these cultures at 5 hours, when BFA is added for detection of CXCL1 by intracellular cytokine staining. Although IL-10 levels can vary between pure WT vs. mixed cultures, the data presented in Figure 6 quantifies KO relative to WT effector production within the same mixed condition and thus at the same IL-10 level, showing a clear effect of IFNAR deficiency on the BMDM response.

Evidence for an IFN-dependent, IL-10 independent effect is shown in Figure 6D for a subset of negative regulatory ISGs. At the same time, consistent with the literature and as conveyed in the Figure 6E model, IL-10 is required for control of specific inflammatory mediators. In agreement with our prior data from IL10R KO BMDM (Figure 6—figure supplement 3), blocking IL10R with antibody as requested prevents shutdown of both TNF and CXCL1 secretion, seen between 6 and 12 hours of stimulation. We now include these blockade data in Figure 6—figure supplement 3. Thus, there are IFN-dependent, IL-10-independent effects on TLR-induced responses, but the impact of IFNAR deficiency on inflammatory cytokine dynamics cannot be evaluated in the absence of IL-10-dependent regulatory mechanisms.

As additional evidence for IFN-mediated negative feedback, we now include data in Figure 6—figure supplement 3 showing defective CXCL1 and TNF shutdown in Ifnar1-/- BMDM, even in the presence of exogenous IL-10. We used 1ng/ml of IL-10, which is approximately 10 times higher than the IL-10 levels detected in WT, *P. aeruginosa-*stimulated BMDM cultures.

b) The results in Figure 2C-D, and Figure 6D and 6F-G are interpreted together to establish the negative regulation of IFN-β independent of IL-10. However, TNF is the readout in Figure 2 and CXCL1 in Figure 6F-G. Since TNF and CXCL1 are observed to show similar "hot spot" behavior under combinations of gram positive and negative bacteria, it would be reasonable to test both of these, particularly in Figure 6F-G.

We have updated Figure 2 to include both TNF and CXCL1 as readout for siRNA and TLR4-dependent shutdown. This includes mRNA time course data showing that shutdown is more rapid and steep for *Cxcl1*, compared to *Tnf.* For Figure 6F-G, we were only able to assess IL-10 dependence for CXCL1 in this assay. As now stately clearly in the text: In contrast to CXCL1, at this 5-8 hour timepoint we did not detect the IFN-dependence for TNF regulation that we routinely seen in 6-12 hour supernatant data, and thus we were not able to compare separate and mixed cultures (Figure 6—figure supplement 4). The more robust *Ifnar1-/-* effect for CXCL1 at this timepoint, compared to TNF, is consistent with our mRNA time course data showing a more rapid shutdown of *Cxcl1* than *Tnf* (Figure 2D).

3) Demonstration of the effects in unstimulated controls.A few experiments/figures would benefit from an untreated control group (usually BMDM that have not been exposed to heat-killed bacteria) to be able to fully interpret the data. This concerns Figures 1A, 3A, 5A, 5B, 6F and Figure 6—figure supplement 3.

We updated the indicated figures to include the level of cytokine detected in BMDM not exposed to heat-killed bacteria, with descriptions of control timing and genotype added to the figure legends. For Figure 1A, 3A, and Figure 6—figure supplement 3, controls for 0-12h unstimulated BMDM are shown as a dashed lines (lower graphs for Figure 1A and 3A). For Figure 5B, a “No bacteria” column has been added for WT, WT+IFNb, and IFNAR KO groups. For Figure 6F, a shaded histogram has been added to the representative flow plots. In some cases these controls are not visible because they are below the level of detection (a zero value), as is now stated in the relevant figure legends. Data in Figure 5A reflect fold change compared to unstimulated, as now clearly stated in the figure legend.